# Lanthanum Promotes Bahiagrass (*Paspalum notatum*) Roots Growth by Improving Root Activity, Photosynthesis and Respiration

**DOI:** 10.3390/plants11030382

**Published:** 2022-01-30

**Authors:** Ying Liu, Juming Zhang

**Affiliations:** 1Qinghai Provincial Key Laboratory of Adaptive Management on Alpine Grassland/Key Laboratory of Superior Forage Germplasm in the Qinghai-Tibetan Plateau, Qinghai Academy of Animal and Veterinary Sciences, Qinghai University, Xining 810016, China; yingliu629@yahoo.com; 2Guangdong Engineering Research Center for Grassland Science, College of Forestry and Landscape Architecture, South China Agricultural University, Guangzhou 510642, China

**Keywords:** lanthanum, bahiagrass, root growth, photosynthesis, respiration

## Abstract

Lanthanum (La), one of the most active rare earth elements, promotes the growth of turfgrass roots. In this study, the mechanisms by which La influences bahiagrass (*Paspalum notatum*) growth were evaluated by the analyses of root growth, root activity, cell wall polysaccharide content, respiration intensity, ascorbic acid oxidase (AAO) and polyphenol oxidase (PPO) activity, the subcellular distribution of mitochondria, transcription in roots, photosynthetic properties, chlorophyll fluorescence parameters, and chlorophyll content. The application of 0.3 mM La^3+^ increased root activity, respiration intensity, AAO activity, and the number of mitochondria in the mature cells of bahiagrass roots. La could significantly improve the net photosynthetic rate, transpiration rate, and chlorophyll fluorescence of bahiagrass. Differentially expressed genes identified by high-throughput transcriptome sequencing were enriched for GO (Gene Ontology) terms related to energy metabolism and were involved in various KEGG (Kyoto Encyclopedia of Genes and Genomes) pathways, including oxidative phosphorylation, TCA (Tricarboxylic Acid) cycle, and sucrose metabolism. These findings indicate that La promotes bahiagrass root growth by improving root activity, photosynthesis, and respiration, which clarifies the mechanisms underlying the beneficial effects of La and provides a theoretical basis for its use in artificial grassland construction and ecological management projects.

## 1. Introduction

Lanthanum (La) belongs to the group of elements known as rare earth elements. Among the fifteen rare earth elements, La is the most active, the most well-studied, and the most widely used. Its application in agricultural production was first proposed in the 1980s. More than three decades of systematic research have generated a wealth of literature indicating that La is a beneficial element for plant growth. In terms of vegetative growth, it can promote seed germination and root growth [1], boost metabolism by the enhancement of photosynthesis [2], and promote the absorption of soil mineral elements by the plant [3]. With respect to reproductive growth, it can improve flower formation and fruit set rates [4]. It can affect the cell skeleton and the function of the plasma membrane [5,6]. The use of La can improve plant resistance to abiotic stresses, such as heavy metals [7], and enhance disease resistance [8]. In short, La is a beneficial nutrient for agricultural production. Recent studies have focused on characterizing the effects of La on plant growth. However, little is known about the mechanisms underlying these effects, including the mechanisms by which La promotes the growth of plant roots, the most important organs for absorbing water and nutrients [9]. 

Bahiagrass (*Paspalum notatum*) is widely used for slope protection and soil and water conservation in southern China owing to its extensive and deep root system. Our preliminary work has shown that La could significantly improve bahiagrass root growth [10]. It has been reported that La absorbed by plant roots is mainly attached to the cell wall of root cells and that La alleviates the toxic effects of aluminum by changing the porosity of the cell wall [11]. Therefore, the objective of this study was to investigate how La promotes the growth of the bahiagrass root system, focusing on its root-promoting mechanism. To achieve this aim, this study was divided into three parts: (1) the evaluation of La’s effect on cell wall components and root activity; (2) the evaluation of the regulatory effects of La on photosynthesis and respiration, as root growth requires a sufficient energy supply; and (3) the evaluation of the genetic basis of the physiological and phenotypic alterations in the root by high-throughput transcriptome sequencing.

## 2. Results 

### 2.1. Measurement of the La Content

Before studying the effect of La on bahiagrass root growth, it is necessary to first determine that the La added to the root nutrient solution can be absorbed by bahiagrass. As shown in Figure 1A, La treatment significantly increased the La content in bahiagrass roots in a time-dependent manner (*p* < 0.05). In addition, the greatest increase was observed after 2 days, and the rate of increase decreased gradually over time. As shown in Figure 1B, La treatment also significantly increased the La content in bahiagrass leaves (*p* < 0.05). Similar to the results obtained for the root tissues, the greatest increase in La was observed on the second day; however, the La content in the leaves did not increase as the treatment time increased (i.e., the increments on day 4 and day 6 were not significant).

### 2.2. Effect of La on Bahiagrass Root Activity

Root activity is the most direct indicator of plant root health. The results of this study showed that La treatment significantly improved the root activity of bahiagrass seedlings (*p* < 0.05) (Figure 2). The application of lanthanum increased root activity nearly fivefold.

### 2.3. Effect of La on the Cell Wall Polysaccharide Content of Bahiagrass Roots

The cell wall polysaccharide content is the most direct indicator of cell wall functionality. La treatment had no significant effect on the cell wall polysaccharide content of bahiagrass roots (*p* > 0.05) (Figure 3). Cell wall polysaccharide content included pectin, hemicellulose I (HCI), and hemicellulose II (HCII). These results suggest that La does not contribute to the regulation of cell wall components, even though La absorbed by the roots mainly accumulates in the cell walls.

### 2.4. Effect of La on Bahiagrass Photosynthesis

#### 2.4.1. Effect of La on the Photosynthetic Properties of Bahiagrass

The net photosynthetic rate is defined as the difference between the rates of photosynthesis and respiration and is often represented as the accumulation of organic matter in the leaf per unit time per unit area. The transpiration rate refers to the amount of transpiration in the leaf per unit time and is one of the main factors affecting the net photosynthetic rate of plants. Stomatal conductance refers to the size of the stomatal opening, which affects the photosynthesis, respiration, and transpiration of plants, and is an important indicator of the impact of stomatal or non-stomatal factors on the photosynthetic rate, along with the intercellular CO_2_ concentration [12,13,14]. Our results showed that La treatment significantly improved the net photosynthetic and transpiration rates of bahiagrass (*p* < 0.05), with no significant effects on stomatal conductance and the intercellular CO_2_ concentration (*p* > 0.05) (Figure 4).

#### 2.4.2. Effect of La on Bahiagrass Chlorophyll Fluorescence Parameters

*F*_v_/*F*_m_ is the maximum quantum yield of a completely open PSII reaction center, reflecting the potential maximum photosynthetic capacity of the plant; *F*_v_′/*F*_m_′ is the excitation light capture efficiency of leaves; ΦPSII refers to the actual quantum yield of PSII with partial closure of the PSII reaction center under light, reflecting the current actual photosynthetic efficiency of the leaves; qP is the photochemical quenching coefficient, reflecting the level of photosynthetic activity of the plant; and ETR refers to the photosynthetic electron transport rate [15,16,17]. La treatment significantly improved the chlorophyll fluorescence parameters, indicating that La effectively promoted photosynthesis in bahiagrass (*p* < 0.05) (Figure 5).

#### 2.4.3. Effect of La on the Chlorophyll Content in Bahiagrass during Photosynthesis 

Chlorophyll is important in the process of photosynthesis, including important roles in electron transport [18,19,20]. However, La treatment did not increase the chlorophyll content in bahiagrass leaves (*p* > 0.05) (Figure 6).

### 2.5. Effect of La on Bahiagrass Respiration

#### 2.5.1. Effect of La on the Respiration Rate of Bahiagrass Roots

Respiration provides a direct energy source for ATP in plant metabolism. The respiration rate is the most direct physiological indicator of respiration. La treatment significantly increased the respiration rate of bahiagrass roots (*p* < 0.05), providing sufficient energy for root growth (Figure 7).

#### 2.5.2. Effect of La on Enzymes Related to Bahiagrass Root Respiration 

Respiration requires a specific complex enzyme reaction system, wherein AAO and PPO enzyme activities are representative physiological indicators. In this study, La treatment significantly increased AAO activity (*p* < 0.05) but had no effect on PPO activity (Figure 8).

### 2.6. Effect of La on Mature Mitochondria in Bahiagrass Roots

The respiration process in plants is completed in mitochondria. In this study, transmission electron microscopy was performed to evaluate mature root cells. The shape and number of mitochondria in nine cells were observed. As shown in Figure 9B,D, La treatment had no effect on mitochondrial morphology. As shown in Figure 9A,C, La treatment increased the number of mitochondria in mature cells. A quantitative analysis revealed that mature La-treated cells, on average, had 3.5 mitochondria, 1.75 more mitochondria than the corresponding number in the control (2 mitochondria).

### 2.7. Effect of La on Transcript Levels

In this study, the total RNA of bahiagrass roots was sequenced using high-throughput transcriptome sequencing technology. Approximately 130 million raw reads were obtained and approximately 110 million clean reads were retained for subsequent analyses after filtering. After assembling the clean reads, approximately 210,000 transcripts were obtained, containing approximately 120,000 potential genes. In total, 24,879 genes were significantly differentially expressed between the La treatment group and control group, of which 15,329 were up-regulated and 9550 were downregulated in the La treatment group. 

A GO analysis of these differentially expressed genes (DEGs) showed that in the biological process category, the DEGs were mainly concentrated in cellular and metabolic processes. In the cellular component category, the DEGs were mainly concentrated in cells, cell parts, and organelles. In the molecular function category, the DEGs were mainly concentrated in binding and catalytic functions (Figure 10). In each major category, the two subcategories where DEGs were concentrated were further refined into subterm classifications. In the cellular and metabolic process category, most of the terms were related to energy metabolism, including enrichment for the electron transport chain, TCA cycle, glycolysis, and other growth-related terms. In the cell and organelle category, there was an enrichment of organelles associated with energy metabolism (mitochondria and chloroplasts). In the binding and catalytic function category, there was an enrichment of binding and enzymes predominantly related to nucleic acids (Table 1).

A KEGG pathway enrichment analysis indicated that DEGs were involved in energy metabolism, oxidative phosphorylation, TCA cycle and sucrose metabolism (Table 2). 

Six DEGs were randomly selected for qRT-PCR validation, and the results are summarized in Figure 11. The qRT-PCR results generally agreed with the high-throughput transcript sequencing results, with a high correlation coefficient of 0.92, indicating that the high-throughput transcript sequencing results were reliable.

## 3. Discussion

### 3.1. Effect of Exogenous La on the La Content in Bahiagrass Leaves and Roots 

La has a relatively high molecular weight. It mostly accumulates around cell walls after it is absorbed by plant roots [11]. However, our results showed that La could also be transported to aboveground parts, although the amount of La absorbed by stems and leaves was limited after a threshold level was reached. Liu et al. obtained similar results when studying cerium, another rare earth element, and pointed out that cerium accumulation is higher in the roots of rice than in the stems and leaves [3]. Our results for La accumulation in leaves provide indirect support for the theory that La treatment in the roots could affect plant physiological processes in aboveground parts. 

### 3.2. Effect of La on the Root Activity and Cell Wall Polysaccharide Content of Roots

We have previously discovered that La could promote root growth in bahiagrass [10]. In this study, we found that La could also increase root activity. Shi et al. and Liu et al. have demonstrated that La promotes the root activity of red bean *(Ormosia microphylla)* [21] and wheat (*Triticum aestivum*) [22]. These results suggest that La enhances the biological function of bahiagrass roots. 

Even though La binds to the root cell wall, many reports have pointed out that La could change the uronic acid content or rigidity of the cell wall [23,24]. We did not detect an effect of La on the polysaccharide content of root cell walls. Similarly, Yang et al. reported that La does not influence the polysaccharide content of rice apical cell walls (pectin, HCI, and HCII) [25]. 

### 3.3. Effect of La on Photosynthesis in Bahiagrass Leaves 

We obtained a comprehensive overview of the effect of La on photosynthesis in bahiagrass. La improved the net photosynthetic rate, transpiration rate, and chlorophyll fluorescence parameters (*F*_v_/*F*_m_, *F*_v_′/*F*_m_′, ΦPSII, qP, and ETR). The increase in the net photosynthetic rate indicated that La treatment could enhance the photosynthetic capacity of bahiagrass. Wen et al. and Wang et al. also reported that low-level La(III) treatment improved photosynthesis in soybean [26] and rice [27], respectively. The observation that La treatment did not increase stomatal conductance or the intracellular CO_2_ concentration indicated that La promoted photosynthesis via the regulation of non-stomatal factors. Given the lack of an increase in stomatal conductance, the significant increase in transpiration rate can be explained by the effect of La on metabolic processes in the whole bahiagrass plant. This might involve the increase in water consumption by plant cells, which reduced the cell potential. The reduced cell potential accelerated water absorption by roots and upward transport, thereby increasing the transpiration rate of the plant. 

Chlorophyll fluorescence is an indicator of photosynthesis. In this study, the enhancement of *F*_v_*/F*_m_, *F*_v_′*/F*_m_′, and ΦPSII indicates that La significantly promoted PSII activity and the primary reaction of photosynthesis as well as the conversion of captured light to chemical energy in leaves. The increase in qP and ETR suggests that La increases the electron transport rate during photosynthesis. Hong et al. pointed out that La could enter spinach chloroplasts and significantly increase the formation of PSII and the electron transport rate [28]. Yan et al. further suggested that 90% of La is present in PSII inside spinach chloroplasts [29]. These results indicate that La could promote the function of PSII and the electron acceptor. Furthermore, in this study, the chlorophyll content in leaves was not increased by La. Shi et al. also showed that La had no significant effect on the chlorophyll content of wheat [21]. Our findings indicate that La raised the electron transport rate in bahiagrass without increasing the chlorophyll content, indicating that La improved the ability of photosynthetic pigments to capture and transport electrons. Hong et al. confirmed that La inside plant chloroplasts could also combine with chlorophyll [28], thereby affecting the electron transport rate during photosynthesis.

### 3.4. Effect of La on Respiration in Bahiagrass Roots

Respiration provides the most direct energy for plant growth and development, including root growth. This study provides a comprehensive overview of the impact of La on the respiration of bahiagrass roots based on the respiration rate, respiration-related enzyme activity, and mitochondrial ultrastructures in mature root cells. Our results showed that La significantly increased the respiration rate of bahiagrass roots, the activity of ascorbic acid oxidase (AAO), and the number of mitochondria in mature cells. This is the first report on the effect of La on the respiration of plant seedlings. Early in the last century, Palmer et al. observed that La could affect human mitochondria [30]. Brooks also reported that an appropriate concentration of La ions could improve the respiration rate of *Bacillus subtilis* [31]. In addition, Hong et al. and Fashui et al. reported that La could promote respiration in rice seeds [32,33]. Overall, our study shows that La clearly influences respiration processes of various taxa. Our research fills the gap regarding the effect of La on respiration in plant seedlings and clearly reveals that La promotes respiration in bahiagrass roots by improving root AAO activity and increasing the number of mitochondria in mature cells.

### 3.5. Effect of La on Transcript Levels in Bahiagrass Roots

In this study, the mechanism by which La promotes bahiagrass root growth was evaluated at the transcriptome level by high-throughput transcriptome sequencing. In the GO enrichment analysis of DEGs, significant enrichment for various functions related to the electron transport chain, chloroplast, and metal ions was observed, providing molecular-level explanations for the promotion of photosynthesis by the combination of La with chlorophyll, the increase in PSII activities, and the efficiency of the electron transport chain at the physiological level. Enrichment of the TCA cycle, glycolysis, mitochondria, inner mitochondrial membrane, and oxidoreductase as well as the significant enrichment of the oxidative phosphorylation, TCA cycle, and sucrose metabolic pathways identified by a KEGG pathway analysis provided further molecular-level support for the physiological-level effect of La, which promotes the respiration of bahiagrass roots by improving AAO activity and increasing the number of mitochondria in mature cells. In addition, the detection of enrichment terms related to cell growth, division, and differentiation in the GO enrichment analysis of DEGs further indicates that La can promote the growth of bahiagrass roots. Finally, quantitative fluorescence PCR verified the reliability of the high-throughput transcriptome sequencing results. 

## 4. Materials and Methods

### 4.1. Plant Materials and Treatment

Bahiagrass seeds were obtained from Clover Group Corporation, Ltd. (Beijing, China). Bahiagrass seed germination and plant growth conditions were the same as those described by Liu et al. [10]. Healthy seeds were germinated on filter paper for 6 days. Seedlings with uniform root lengths were selected. The concentration of La^3+^ (LaCl_3_) was 0.3 μΜ. The treatment duration was 6 days. The containers were placed in a growth chamber at 28 °C, 80% relative humidity, 650 µmol m^−2^ s^−1^ photosynthetic active radiation, and a photoperiod of 12 h. La treatment and the control were in the same condition.

### 4.2. Elemental Analysis

Fresh plants were collected and separated into roots and leaves. They were then washed thoroughly with deionized water, and a microwave-assisted digestion procedure was applied. Approximately 0.1 g of sample (FW) was weighed into Teflon bombs. Then, 5 mL of HNO_3_ (70%) and 2 mL of H_2_O_2_ (30%) were added, followed by digestion. The digestion adopted the fractional stepwise temperature raising method. The temperature was heated to 120 °C for 5 min, then to 150 °C for 5 min, and then to 180 °C for 5 min. After digestion, the samples were transferred into polypropylene tubes and filled to 10.0 cm^3^. Subsequently, the La content was investigated by inductively coupled plasma mass spectrometry (ICP-MS; Agilent, Hachioji, Japan).

### 4.3. Root Activity

Root activity was measured using the modified triphenyl tetrazolium chloride (TTC) method [34,35]. TTC is mainly succinate dehydrogenase, which is reduced by dehydrogenases. Dehydrogenase was expressed as the quantity of deoxidized TTC, which is an index of root activity. In brief, 50 fresh root tips (1 cm) were immersed in 10 mL of an equally mixed solution of TTC (0.4%) and phosphate buffer (0.1 mol·L^−1^, pH 7.0) and kept in the dark for 3 h at 37 °C. Subsequently, 2 mL of H_2_SO_4_ (1 mol·L^−1^) was added to stop the reaction. The immersed root tips were dried with filter paper and extracted with ethyl acetate. The extracted solution was transferred into a tube with ethyl acetate cleaning solution to a total volume of 10 mL, and absorbance was read at 485 nm. Root activity = amount of TTC reduction (µg)/100 fresh root tips (root) × time (h).

### 4.4. Cell Wall Polysaccharide Content

Cell walls were isolated from the upper 1 cm of root tips of 6-day-old seedlings of bahiagrass according to the procedure described by Zhong and Lauchli [36]. After freeze-drying, the cell wall materials were fractionated into three fractions: pectin, HCI, and HCII. The pectin fraction was extracted twice with 0.5% ammonium oxalate buffer containing 0.1% NaBH_4_ (pH 4) in a boiling water bath for 1 h, and the supernatants were pooled. Pellets were subsequently subjected to triple extraction with 4% KOH containing 0.1% NaBH_4_ at 25 °C for 24 h, followed by extraction with 24% KOH containing 0.1% NaBH4. The pooled supernatants from the 4% and 24% KOH extraction yielded the HCI and HCII fractions, respectively. 

The uronic acid content was assayed according to the method of Blumenkrantz and Asboe-Hansen [37] for each cell wall fraction. Galacturonic acid (GalA) was used as a calibration standard, and the root pectin content is expressed as GalA equivalents.

### 4.5. Photosynthetic Characteristics

Photosynthetic properties (net photosynthetic rate, transpiration rate, stomatal conductance, and intercellular CO_2_ concentration) were measured using a portable photosynthesis system (Li-6400; Li-Cor Inc., Lincoln, NE, USA). Measurements were carried out in a 2 cm^2^ leaf area with an airflow rate of 500 µmol s^−1^. The photon flux density (PPFD) was 1500 µmol m^−2^ s^−1^, and the leaf temperature was controlled at 25 °C. The characteristic parameters evaluated were the net photosynthetic rate, transpiration rate, stomatal conductance, and intercellular CO_2_ concentration. 

### 4.6. Chlorophyll Fluorescence Parameters

After a seedling adaptation period of 30 min in the dark under a controlled temperature (25 °C) and treatment for 6 days with La, the minimum fluorescence (*F*_o_), maximum fluorescence (*F*_m_), constant fluorescence (*F*_s_), maximum fluorescence with actinic light (*F*_m_′), minimum fluorescence with far-red light (*F*_o_′), effective photochemical quantum yield adapted to light (ΦPSII), photochemical quenching coefficient (qP), and electron transport rate (ETR) were measured with a portable chlorophyll fluorometer (PAM-2000; WALZ, Effeltrich, Germany) using a leaf-clip holder (2030-B; Walz, Germany). The photon flux density (PPFD) was 1500 µmol m^−2^ s^−1^, and the actinic light treatment time was 30 min. Other parameters were calculated as *F*_v_/*F*_m_ = (*F*_m_ − *F*_o_)/*F*_m_ and *F*_v_′/*F*_m_′ = (*F*_m_′ − *F*_o_′)/*F*_m_′.

### 4.7. Chlorophyll Content

Fresh leaves were weighed and 0.1 g was obtained after treatment for 6 days. Samples were placed into a 25 mL scale test tube after shearing. Then, 80% acetone was added and the leaves were carefully washed and adhered to the wall of the bottle. The bottle was covered and kept in a dark box overnight and shocked three times. After filtration or centrifugation, the chloroplast pigment extract was poured into a colorimetric cup with a light diameter of l cm. Absorbance was measured at wavelengths of 663 nm and 645 nm with 80% acetone as a blank control. The chlorophyll content was calculated using the following equations: chlorophyll (C) = 20.2D645 + 8.02D633 [38] and chlorophyll content (mg/g) = (chlorophyll concentration C × extracted liquid volume × dilution factor)/sample fresh weight.

### 4.8. Root Respiration Rate

After 6 days of treatment, bahiagrass root (approximately 0.2 g fresh weight) was taken and immediately placed into an oxygen suction tank containing 8 mL of reaction medium. The Clark oxygen electrode (YSI 5300, Yellow Springs, OH, USA) was used to measure root respiration rate, and the control and treatment were repeated three times each.

### 4.9. Respiration-Related Enzyme Activity

The enzyme activities of two respiration-related enzymes, ascorbic acid oxidase (AAO) and polyphenol oxidase (PPO), were measured. The method used to determine AAO enzyme activities was described by Oberbacher and Vines [39]. PPO enzyme activity was determined according to Jiang [40].

### 4.10. Observation of the Mitochondrial Ultrastructure in Mature Root Cells

The mitochondrial ultrastructure of mature cells in bahiagrass roots was observed by transmission electron microscopy (TEM; Tecnai 12, FEI, Eindhoven, The Netherlands). After treatment for 6 days, 10 plant roots were randomly selected for sample preparation for each treatment, three samples were randomly selected for slicing at the end of sample preparation, and three cells were randomly selected for observation. Root samples were prepared according to the method described by Xu et al. [41]. 

### 4.11. Transcriptome Sequencing

Total RNA was extracted from root tissues using TRIzol reagent (Life Technologies, Carlsbad, CA, USA) according to the manufacturer’s protocol. RNA degradation and contamination were detected by 1% agarose gel electrophoresis. RNA purity was checked using a kaiaoK5500^®^ spectrophotometer (Kaiao, Beijing, China). RNA integrity and concentration were assessed using the RNA Nano 6000 Assay Kit and the Bioanalyzer 2100 system (Agilent Technologies, Santa Clara, CA, USA). A total of 3 μg of RNA per sample was used as the input material for RNA sample preparation. Sequencing libraries were generated using a NEBNext^®^ Ultra™ RNA Library Prep Kit for Illumina (#E7530L; NEB, Ipswich, MA, USA) following the manufacturer’s recommendations, and index codes were added to attribute sequences to each sample. After cluster generation, the libraries were sequenced on the Illumina HiSeq 2500 platform to generate 50 bp single-end reads. De novo transcriptome assembly was performed using Trinity (http://trinityrnaseq.sf.net/). RPKM values were used to normalize transcript levels [42]. Fold change values of >2 indicated significant differences in gene expression between treatments. Unigenes were identified using TransDecoder (version r2013_08_14) [43] and were functionally annotated using the Blast2GO Gene Ontology (GO) functional annotation suite (E-value < 10^−5^) (http://www.blast2go.org/). Metabolic pathways were predicted using Kyoto Encyclopedia of Genes and Genomes (KEGG) mapping.

### 4.12. Quantitative Real-Time PCR Analysis

A total of 1–2 μg of RNA was reverse-transcribed using the TaKaRa PrimeScript II 1st Strand cDNA Synthesis Kit (D6210A) according to the manufacturer’s instructions. qRT-PCR was performed using a Bio-Rad CFX96 Real-Time PCR System (Bio-Rad, Hercules, CA, USA). TaKaRa SYBR Premix Ex Taq II (Perfect Real Time) was used for amplification. The β-tubulin sequence was used as an internal control to measure the relative transcript levels. Information on the oligonucleotide sequences used for qRT-PCR is provided in Table 3.

### 4.13. Statistical Analysis

Variance analyses were performed using SPSS (version 13.0; Chicago, IL, USA). Differences between the means of the treatments for each parameter were assessed using the least significance difference test (LSD) at *p* = 0.05. 

## 5. Conclusions

La promoted the growth of bahiagrass roots mainly by improving root activity, photosynthesis (i.e., improving net photosynthetic rate, transpiration rate, chlorophyll fluorescence, PSII activity, electron transport chain efficiency, and oxidative phosphorylation), and respiration (i.e., increasing respiration intensity and the number of mitochondria in mature cells, enhancing AAO activity, TCA cycle, and sucrose metabolism), thereby providing energy for root growth. These results explain the reason why lanthanum promoted bahiagrass root growth to some extent and could provide a theoretical basis for using lanthanum to rapidly establish bahiagrass artificial slope protection grassland and other ecological management projects.

## Figures and Tables

**Figure 1 plants-11-00382-f001:**
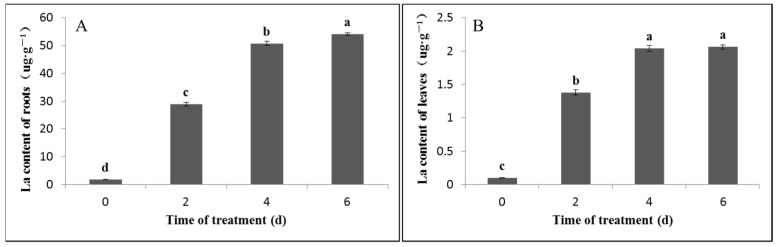
La content in bahiagrass roots and leaves for various La treatment durations. Note: Bars with different letters are significantly different at *p* = 0.05. (**A**) La content of roots, (**B**) La content of leaves.

**Figure 2 plants-11-00382-f002:**
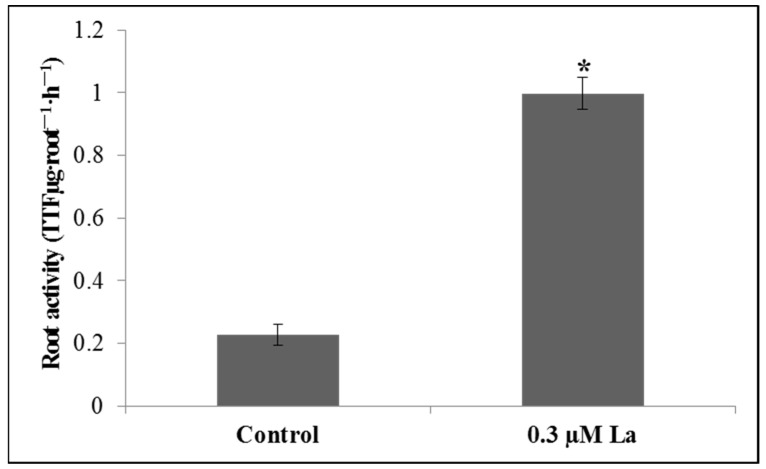
Effect of La on root activity of bahiagrass roots. Note: Asterisks (*) indicate significant differences at *p* = 0.05.

**Figure 3 plants-11-00382-f003:**
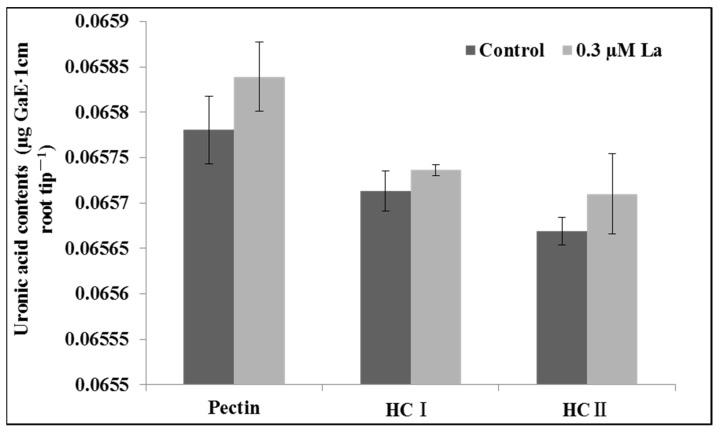
Effect of La on the uronic acid content of bahiagrass.

**Figure 4 plants-11-00382-f004:**
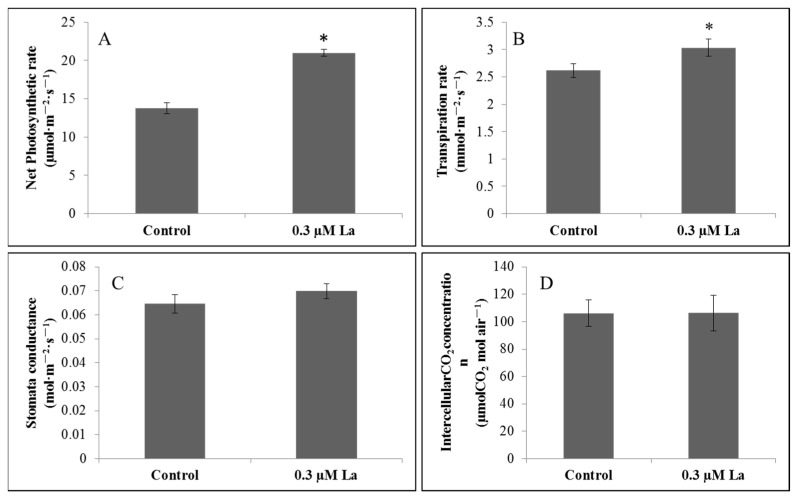
Effect of La on photosynthetic properties of bahiagrass leaf. Note: Asterisks (*) indicate significant differences at *p* = 0.05. (**A**) Net Photosynthesis rate, (**B**) Transpiration rate, (**C**) Stomatal conductance, (**D**) intercellularCO_2_ concentration.

**Figure 5 plants-11-00382-f005:**
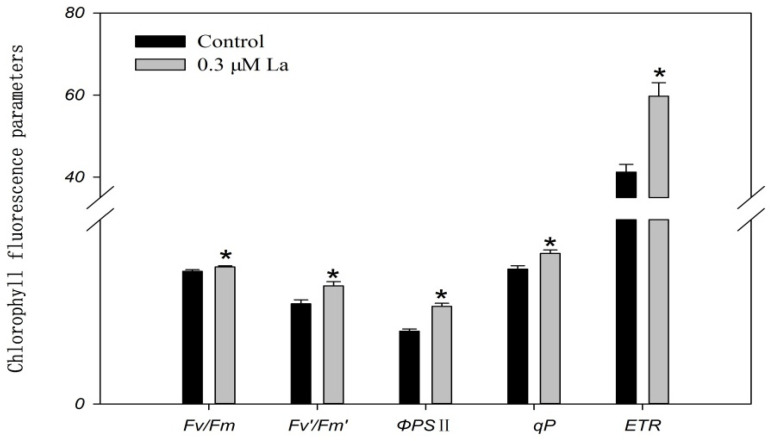
Effect of La on chlorophyll fluorescence parameters in bahiagrass Note: Bars within the same parameter with asterisks are significantly different at *p* = 0.05.

**Figure 6 plants-11-00382-f006:**
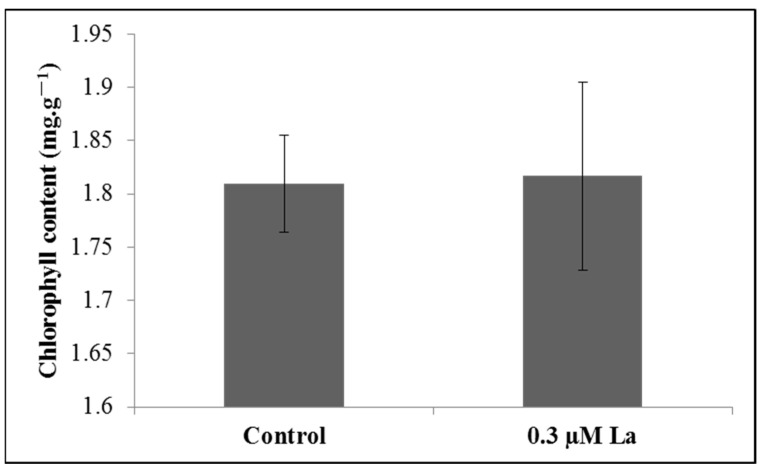
Effect of La on the chlorophyll content of bahiagrass.

**Figure 7 plants-11-00382-f007:**
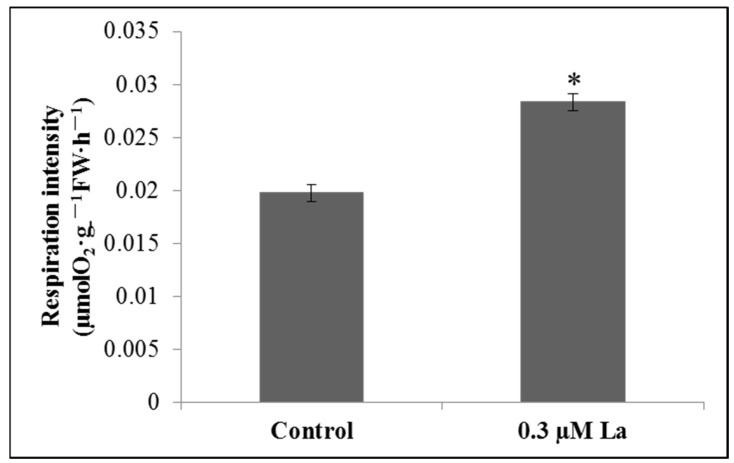
Effect of La on respiration intensity of bahiagrass roots. Note: Asterisks (*) indicate significant differences at *p* = 0.05.

**Figure 8 plants-11-00382-f008:**
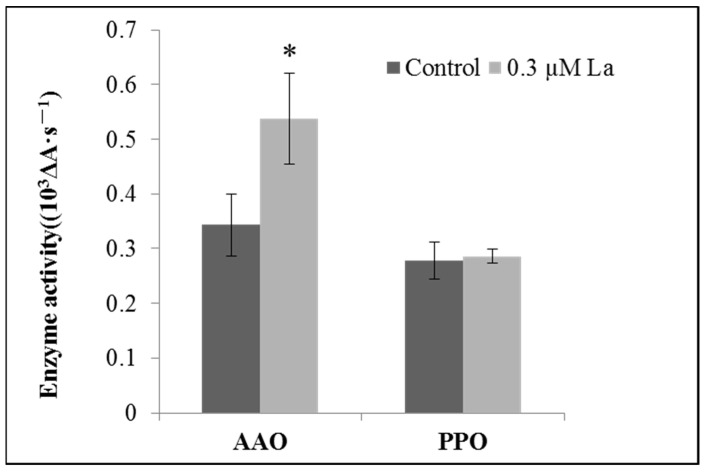
Effect of La on ascorbic acid oxidase (AAO) and polyphenol oxidase (PPO) activity in bahiagrass roots. Note: Bars within the same enzyme with asterisks are significantly different at *p* = 0.05.

**Figure 9 plants-11-00382-f009:**
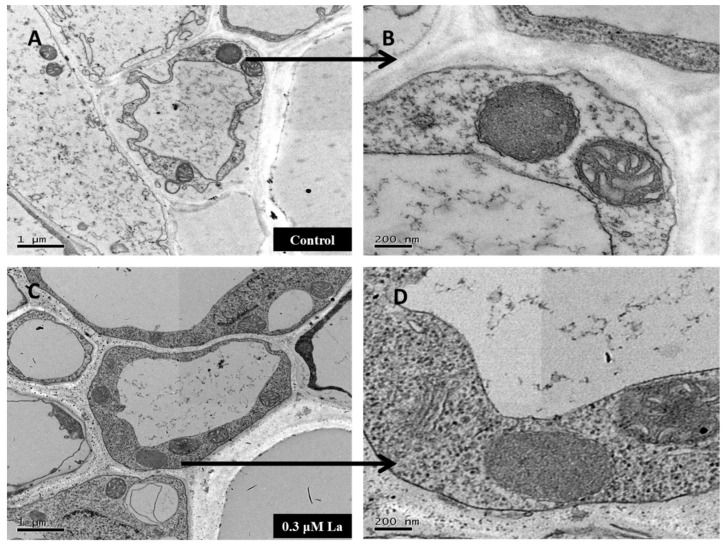
Effect of La on the subcellular distribution of mitochondria in bahiagrass mature root cells. (**A**) Mitochondrias in bahiagrass mature control root cells (Bar =1 μm), (**B**) Individual mitochondria in bahiagrass mature control root cells (Bar =200 nm), (**C**) Mitochondrias in bahiagrass mature treatment (0.3 μM La) root cells (Bar =1 μm), (**D**) Individual mitochondria in bahiagrass mature treatment (0.3 μM La) root cells (Bar =200 nm).

**Figure 10 plants-11-00382-f010:**
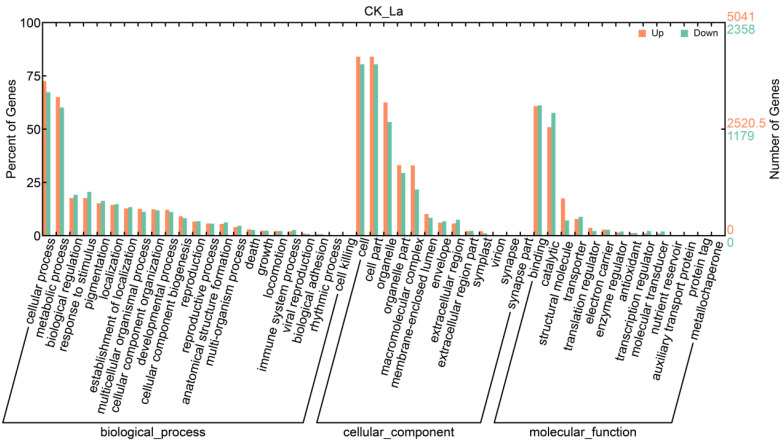
Gene ontology (GO) classification of differentially expressed genes. Note: The *x*-axis indicates the subterm within each main term. The left *y*-axis indicates the percentage of a specific term for differentially expressed genes in each main term. The right *y*-axis represents the number of differentially expressed genes.

**Figure 11 plants-11-00382-f011:**
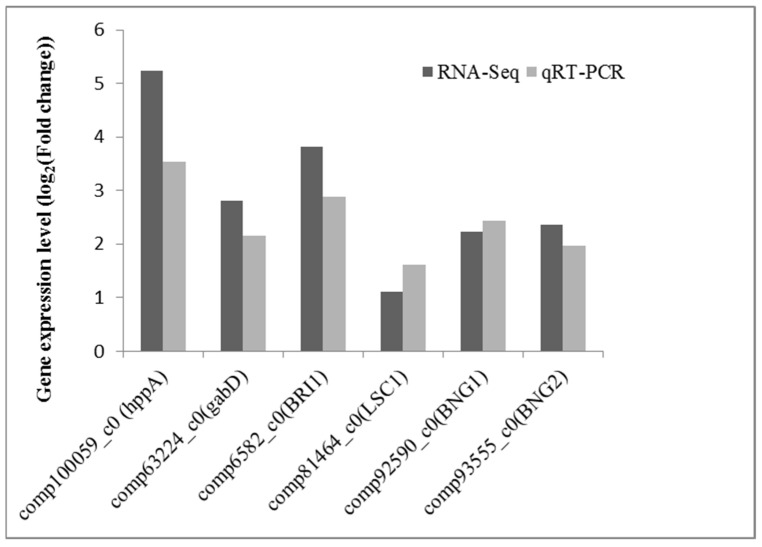
Verification of RNA-seq results by real-time quantitative PCR (qRT-PCR).

**Table 1 plants-11-00382-t001:** Gene ontology (GO) subterm classification of differentially expressed genes.

GO Term	GO Subterm	Up Count	Percent	Down Count	Percent
biological_process	electron transport chain	95	0.018845	42	0.017812
biological_process	ATP catabolic process	88	0.017457	43	0.018236
biological_process	GTP catabolic process	80	0.01587	19	0.008058
biological_process	tricarboxylic acid cycle	76	0.015076	42	0.017812
biological_process	growth	60	0.011902	10	0.004241
biological_process	glycolysis	57	0.011307	23	0.009754
biological_process	carbohydrate metabolic process	53	0.010514	30	0.012723
biological_process	cell division	50	0.009919	17	0.00721
biological_process	ATP biosynthetic process	44	0.008728	15	0.006361
biological_process	cell differentiation	38	0.007538	23	0.009754
cellular_component	cytoplasm	774	0.153541	330	0.139949
cellular_component	nucleus	635	0.125967	325	0.137829
cellular_component	ribosome	504	0.09998	90	0.038168
cellular_component	plasma membrane	467	0.09264	257	0.108991
cellular_component	cytosol	354	0.070224	188	0.079729
cellular_component	mitochondrion	301	0.05971	134	0.056828
cellular_component	nucleolus	248	0.049197	60	0.025445
cellular_component	extracellular region	145	0.028764	105	0.044529
cellular_component	chloroplast	134	0.026582	53	0.022477
cellular_component	mitochondrial inner membrane	130	0.025789	54	0.022901
molecular_function	ATP binding	1012	0.200754	518	0.219678
molecular_function	metal ion binding	582	0.115453	286	0.121289
molecular_function	GTP binding	217	0.043047	57	0.024173
molecular_function	nucleotide binding	212	0.042055	94	0.039864
molecular_function	GTPase	146	0.028963	40	0.016964
molecular_function	electron carrier	138	0.027376	62	0.026293
molecular_function	protein serine/threonine kinase	136	0.026979	125	0.053011
molecular_function	translation elongation factor	117	0.02321	17	0.00721
molecular_function	ATPase	89	0.017655	52	0.022053
molecular_function	oxidoreductase	74	0.01468	51	0.021628

**Table 2 plants-11-00382-t002:** Enriched KEGG metabolic pathways for differentially expressed unigenes.

Name	Gene_in_DE (Number)	Gene_in_Background (Number)	*p*	q	Result
Oxidative phosphorylation	181	602	0.000388	0.004069	yes
Starch and sucrose metabolism	59	326	0.010324	0.041963	yes
Citrate cycle (TCA cycle)	114	368	0.010918	0.04299	yes

Note: q < 0.05 means significant difference. DE means differential expression.

**Table 3 plants-11-00382-t003:** Primer sequences for qRT-PCR.

Gene Name	Forward Primer (5′-3′)	Reverse Primer (5′-3′)
comp100059_c0 (hppA)	TTCCTGACTGCTGAGGGAGT	AATCGAGACGAGAGCGAATC
comp63224_c0(gabD)	GCAAGATCAGTGCTGCTGAG	GGTTTCCCACCTCGTCATTA
comp6582_c0(BRI1)	CATCCCTTGGCATTCTCACT	CTTCAGCTCTGTGCAGCTTG
comp81464_c0(LSC1)	AGAGGAGGAGGGACGAAGAG	TCATCTACCAGGGCTTCACC
comp92590_c0(BNG1)	ACCGTCAGGAGCACAAAGAT	ACAAGGGTGACCGAGAAATG
comp93555_c0(BNG2)	GGGAGTTAATGCGTCGAGAA	CAGAGAGGCCGGCATATAAA
β-tubulin	GTGGAGTGGATCCCCAACAA	AAAGCCTTCCTCCTGAACATGG

## Data Availability

The data presented in this study are available upon request from the corresponding author. The data are not publicly available due to the ongoing downstream studies based on this study.

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
