# Peer review of "Lanthanum Promotes Bahiagrass (Paspalum notatum) Roots Growth by Improving Root Activity, Photosynthesis and Respiration"

_plants, 2022, doi:10.3390/plants11030382_

Round 1

Reviewer 1 Report

The work presented for review has an interesting idea.

The reviewer's doubts are raised by the sense of using a rare element, especially since LaCl3 is very expensive.

 Also introducing it into the soil, will it not have a negative impact on the environment and, for example, animals??

Line 35/35 – this is a paragraph?

lines 80, 202 and 297 – I would suggest using an abbreviation to connect with a previously developed word and decide on one form.

Line 188 – style analysis

Line 218 – I think the discussion scheme has been demolished

References to documents are missing from the text.

The text contains many editing errors.

Author Response

Response to Reviewer 1 Comments

The manuscript was carefully revised following the comments and suggestions of you. Specific responses to each individual comment are listed below. All comments and suggestions were taken into consideration in the revision. The original comments were included for reference.  We strongly believe that the manuscript was improved substantially as the results of the constructive review and our careful response to each comment.

Point 1: l. 17:  Subscrip

Response 1: corrected

Point 2: l. 41:  add “which”

Response 2: added

Point 3: l. 36: please correct

Response 3: corrected

Point 4: l. 50:  ref?

Response 4: added

Point 5: l. 52-56:  This part is to confusing, please simplify

Example:

To achieve this aim, this study was divided into 3 parts: (1) Evaluation of La's effect on cell….; (2) Evaluation of the regulatory effects of La on…., as root growth requires suficiente energy supply, and (3) Evaluation of the genetic basis…. by high-throughput...

Response 5: simplified to “To achieve this aim, this study was divided into 3 parts: (1) Evaluation of La's effect on cell wall components and root activity were evaluated; (2) Evaluation of the regulatory effects of La on photosynthesis and respiration, as root growth requires suficiente energy supply, and (3) Evaluation of the genetic basis of physiological and phenotypic alterations in the root by high-throughput transcriptome sequencing.”

Point 6: l. 76:  must be together with the Fig

Response 6: corrected

Point 7: l. 80:  there is a statistic test for this? Please provide this information.

Response 7: added

Point 8: l. 82:  where is this results? Figure 3? Must refer it in the text.

Response 8: Figure 3 was the results. We added it in the text

Point 9: l. 88-93:  ref?

Response 9: added

Point 10: l. 98:  figure?? Must always refer the Figure supporting the results.

Response 10: added

Point 11: l. 109:  ref?

Response 11: added

Point 12: l. 110:  Figure?

Response 12: added

Point 13: l. 116:  ref?

Response 13: added

Point 14: l. 117:  Figure 6?

Response 14: added

Point 15: l. 125:  Figure?

Response 15: added

Point 16: l. 132:  Figure?

Response 16: added

Point 17: l. 140:  space

Response 17: corrected

Point 18: l. 144:  data non shown

Response 18: added

Point 19: l. 228:  rise.

Response 19: changed ”increased” to “rose”

Point 20: l. 244:  remove "in conclusion", maybe "overall, our study shows that La clearly…"

Response 20: revised to “Overall, our study shows that La clearly influences respiration processes of various taxa.”

Point 21: l. 272: What about the control? same condition without La

Response 21: added sentence “La treatment and the control were in the same condition.”

Point 22: l. 279: specify the condition for the digestion

Response 22: added sentence “The digestion adopted the fractional stepwise temperature raising method. The temperature was heated to 120°C for 5 min, then to 150°C for 5 min, and then to 180°C for 5 min.”

Point 23: l. 382: where is the statistic section? It was performed, so must be included in the material and methods.

Response 23: added one section “4.13. Statistical analysis    Variance analyses were performed using SPSS (version 13.0; Chicago, IL, USA). Dif-ferences between means of the treatments for each parameter were assessed using the least significance test (LSD) at p=0.05.”

Point 24: l. 395: remove the space

Response 24: removed

Reviewer 2 Report

Dear authors,

Thank you for submitting this well-designed research work, which I consider to be very rich in terms of methodology and results. The chemical, biological, biochemical and molecular approach presented here, makes this work a good example of thorough investigation on any subject. In addition, the topic addressed here is of great relevance in plant ecology.

However, I am living some comments that I believe will be helpful  for an improved presantation of your work.

Best regards

Author Response

Response to Reviewer 1 Comments

The manuscript was carefully revised following the comments and suggestions of you. Specific responses to each individual comment are listed below. All comments and suggestions were taken into consideration in the revision. The original comments were included for reference.  We strongly believe that the manuscript was improved substantially as the results of the constructive review and our careful response to each comment.

Point 1: Also introducing it into the soil, will it not have a negative impact on the environment and, for example, animals??

Response 1: Lanthanum in low concentrations is not harmful to the environment or animals. Under the premise of good concentration control, lanthanum has a greater role in promoting environmental protection and agricultural production. Its application in agricultural production was first proposed in the 1980s. So far, there have been no reports of harm to the environment or animals at low concentrations. In this study, the concentration of lanthanum we used was very low and was selected from a list of five concentrations to best promote plant growth.

Point 2: Line 35/35 – this is a paragraph?

Response 2: corrected

Point 3:  lines 80, 202 and 297 – I would suggest using an abbreviation to connect with a previously developed word and decide on one form.

Response 3: We have used one form abbreviation now.

Point 4:  Line 188 – style analysis

Response 4: We deleted the redundant “after”.

Point 5:  Line 218 – I think the discussion scheme has been demolished

Response 5: We revised the sentence to “Hong et al. confirmed that La inside plant chloroplasts also could combine with chloro-phyll [20]”. Our results showed that La improved the net photosynthetic rate, transpiration rate, and chlorophyll fluorescence parameters (Fv/Fm, Fv′/Fm′, ΦPSII, qP and ETR). The enhancement of Fv/Fm, Fv'/Fm' and ΦPSII indicates that La significantly promoted PSII activity and the primary reaction of photosynthesis as well as the conversion of captured light to chemical energy in leaves. The increase in qP and ETR suggests that La increases the electron transport rate during photosynthesis. In this study, we cannot determine how much of lanthanum is combined with PS2 or chlorophyll. According to our experimental results, we think that part of lanthanum is combined with PS2 and part with chlorophyll.  Those literatures support this diagnosis. 

Point 6:  References to documents are missing from the text.

Response 6: added

Point 7:  The text contains many editing errors.

Response 7: The whole manuscript was carefully revised and all modifications were marked in red.

Round 2

Reviewer 1 Report

The revisions have improved the publication and I wish the authors success in their continued research.